# Integration of Farm Financial Accounting and Farm Management Information Systems for Better Sustainability Reporting

Krijn Poppe [1] , Hans Vrolijk [1,\*] and Ivor Bosloper [2]

1   Wageningen Economic Research, Wageningen University and Research, 2595 BM The Hague, The Netherlands
2   Dacom, 9752 HR Haren, The Netherlands
\*   Correspondence: hans.vrolijk@wur.nl

**Abstract:** Farmers face an increasing administrative burden as agricultural policies and certification systems of trade partners ask for more sustainability reporting. Several indicator frameworks have been developed to measure sustainability, but they often lack empirical operationalization and are not always measured at the farm level. The research gap we address in this paper is the empirical link between the data needs for sustainability reporting and the developments in data management at the farm level. Family farms do not collect much data for internal management, but external demand for sustainability data can partly be fulfilled by reorganizing data management in the farm office. The Farm Financial Accounts (FFAs) and Farm Management Information Systems (FMISs) are the main data sources in the farm office. They originate from the same source of note-taking by farmers but became separated when formalized and computerized. Nearly all European farms have a bank account and must keep financial accounts (e.g., for Value-Added Tax or income tax) that can be audited. Financial accounts are not designed for environmental accounting or calculating sustainability metrics but provide a wealth of information to make assessments on these subjects. FMISs are much less frequently used but collect more technical and fine-grained data at crop or enterprise level for different fields. FMISs are also strong in integrating sensor and satellite data. Integrating data availability and workflows of FFAs and FMISs makes sustainability reporting less cumbersome regarding data entry and adds valuable data to environmental accounts. This paper applies a design science approach to design an artifact, a dashboard for sustainability reporting based on the integration of information flows from farm financial accounting systems and farm management information systems. The design developed in this paper illustrates that if invoices were digitized, most data-gathering needed for external sustainability reporting would automatically be done when the invoices is paid by a bank transfer. Data on the use of inputs and production could be added with procedures as in current FMISs, but with less data entry, fewer risks of differences in outcomes, and possibilities of cross-checking the results.

**Keywords:** sustainability reporting; farm accounting; farm management systems; design science; digitization

## 1. Introduction

Farmers have not chosen their profession to work in the farm office and spend their time on administrative burdens. However, farms, especially their relations with food processors and governments, have become more data-intensive. Food processors and retailers started to certify farms on food safety management (GlobalGap) as a reaction to food safety crises in the 1990s [1]. More recently, this has developed into sustainability schemes where farmers must provide evidence of their sustainability performance [2]. Governments have started to ask for data in the Common Agricultural Policy to monitor the conditionality requirements of direct payments and eco-schemes [3]. The availability and quality of data are also of increasing importance in environmental assessments and

life cycle analysis; see, for example, [4–6]. This trend is expected to increase with more stringent sustainability, climate, and biodiversity policies. The EU's Corporate Sustainability Directive, which requires large companies to disclose information on the sustainability of their suppliers (scope-3), and the Farm Sustainability Data Network, as proposed in the Farm to Fork strategy [7], illustrate this. To fulfill the sustainability monitoring and reporting needs, several authors have developed indicator frameworks to measure the different dimensions of sustainability [8–11]. These indicator frameworks are often on a conceptual level without an empirical operationalization or based on ad hoc data collection, are measured on different levels of aggregation, and are not harmonized among countries or applications [12].

As farms get larger, operate more fields or have more animals, and adopt precision farming technologies, the role of data in farm management also increases [13–16]. Precision farming technologies not only demand data (e.g., prescription maps to generate instructions for spraying machinery), but the machines also generate large amounts of data, e.g., on inputs used or harvested products at a detailed geographical scale. The need for more internal data management is sometimes solved by adopting Enterprise Resource Planning software (ERP). ERP systems have been designed as standardized software packages that combine the functionality of multiple business functions into one integrated system [17,18]. In practice, their use is concentrated on very large farms such as in horticulture [19] or large arable farms in Central and Eastern Europe that are organized as companies, with pay-roll accounting and invoicing processes. Many of these companies have their own internal bookkeeper. A much lighter type of ERP for family farms, with mainly self-employment and less need for internal management information, is lacking. Such farms receive all their invoices (also on sales to a cooperative) and do their data recording mainly for external purposes, and even their Farm Management Information System is mainly used for the data needs of food processors. In addition, current ERP systems are not very responsive to big data developments [20].

The growing importance of data on the farm, driven by internal but especially external developments, poses new demands for data management in the farm office to make use of current digital technologies. Many farmers currently use two systems for the organization of their data: Farm Financial Accounting (FFA), which is an application of conventional business accounting, and a Farm Management Information System (FMIS), a software system to record the day-to-day activities at the farm [21,22]. These two systems developed from the same source.

Early farm accounts that historians have recovered are diaries in which a farmer notes the things that happen on a farm (see, for example, [23]). Many farmers still remember how older generations made notes and sketches of a cropping plan in a diary format, which is fully in line with the advice of Luca Pacioli, the Italian Renaissance scholar who was the first to describe double-entry accounting [24,25]. He advised [26] a day book (or memorandum book) for merchants: "The memorandum book, or, according to others scrap book or blotter, is a book in which the merchant shall put down all his transactions, day by day, hour by hour. In this book he will put down in detail everything that sells or buys, and every transaction without leaving a jot: who, what, when, were, mentioning everything to make it fully as clear (..)". Although double-entry accounting was seen as theoretically superior [27], the practice of a day book and single-entry accounting was the reality in agriculture—if books were kept at all.

Farm accounting became more formalized when income taxes were introduced in many European countries between 1850 and 1950 [28]. That led to a formal cash book in addition to the diary. The cash book was filled in by the farmer to enable a specialized accountant to calculate income and profits to be declared to the tax authorities. Extension services created field books (and animal recording books) that were another formalized split-off from the diary; field books are still used in extension projects (i.e., [29]). The field books were computerized in FMISs with the introduction of the personal computer in the



1980s. Cash books were replaced by binders with bank payment papers, and accountants started to use accounting software on mainframes already in the 1970s.

Farm Management Information Systems (FMISs) in agriculture have evolved from simple farm recordkeeping into sophisticated and complex systems to support production management (see [30–33] for reviews of the role and functionalities of farm management systems). The purpose of current FMISs is to meet the increased demands to reduce production costs, comply with agricultural standards, and maintain high product quality and safety [34]. Melzer and Gandorfer [32] also conclude that automated monitoring and recording of farm processes and production materials to comply with legislative standards is highly relevant for farmers. The use of FMISs helps to keep track of what has been done to (1) plan future operations and (2) automate reporting for different purposes.

The preceding shows that FFAs and FMISs are historically related accounting techniques that have been separated over time for practical reasons related to sharing data with accountants or extensionists. They have both developed and were digitalized but did not necessarily reduce administrative tasks in the farm office. Some attempts have been made to link the functionalities of FMISs and FFAs. Carli and Canavari [35] present a model to support the use of Direct Costing and Activity Based Costing methodologies in farm management information systems. Halabi and Carroll [36] take the accountant's perspective and analyze how to increase the value of farm financial information for farm management. Outside of agriculture in large organizations and also for some large agricultural companies, Enterprise Resource Planning software (ERP systems like SAP) have integrated data [37].

The research gap we address in this paper is the empirical link between the data needs for sustainability reporting and the developments in data management at the farm level. One branch of scientific literature describes a broad range of indicator frameworks [8–11], while data gathering and reporting implications are often not considered [12]. Another branch of literature defines and analyses the functionality of FFAs [28] and FMISs [30–33] for data management at the farm level, with a strong focus on the primary objectives of these separate systems. Farmers face the problem of using data from these systems to report their sustainability performance. Better use of digital technologies could help solve these issues [38]. This paper focuses on the less studied link between data needs for sustainability indicators and the data available within the farm.

Given this research gap, the objective of this paper is to develop an approach to fulfill sustainability reporting needs in such a way that it limits the administrative burden, is as user-friendly as possible, makes use of the available data, and provides mechanisms to assure the quality of the sustainability indicators.

## 2. Materials and Methods

This paper builds upon the principles of design science. Design science is an interdisciplinary approach combining design principles with the scientific method to develop new solutions to complex problems. Design science was inspired by Simon [39] and is a well-accepted research paradigm in information systems research [40,41]. Design science focuses on building purposeful artifacts that address previously unsolved problems and are evaluated with respect to the utility provided in solving those problems [42,43]. Design science tries to extend the boundaries of human and organizational capabilities by creating new and innovative artifacts. Artifacts are often material products (a new product such as a vase or a car) but can also be software, a service, or a new business concept. Artifacts are presented in a structured form that may vary from formal logic and rigorous mathematics to software or informal natural language descriptions. The design phases, the protocol, and the methods and materials used in this study are shown in Figure 1.

Designing is a creative process that benefits from a well-organized approach. The UK's Design Council [44] suggests breaking down the creative design process into four phases: Discover (insight into the problem), Define (the area to focus upon), Develop (potential solutions), and Deliver (solutions that work). Design principles can help in

creating (Develop) and evaluating alternative solutions. Especially in the Develop phase, creative methods and tools (e.g., graphic modeling, as we apply in this paper) can be useful. Reviews of design science studies reveal that there is wide variation in the steps and methods applied [45,46].

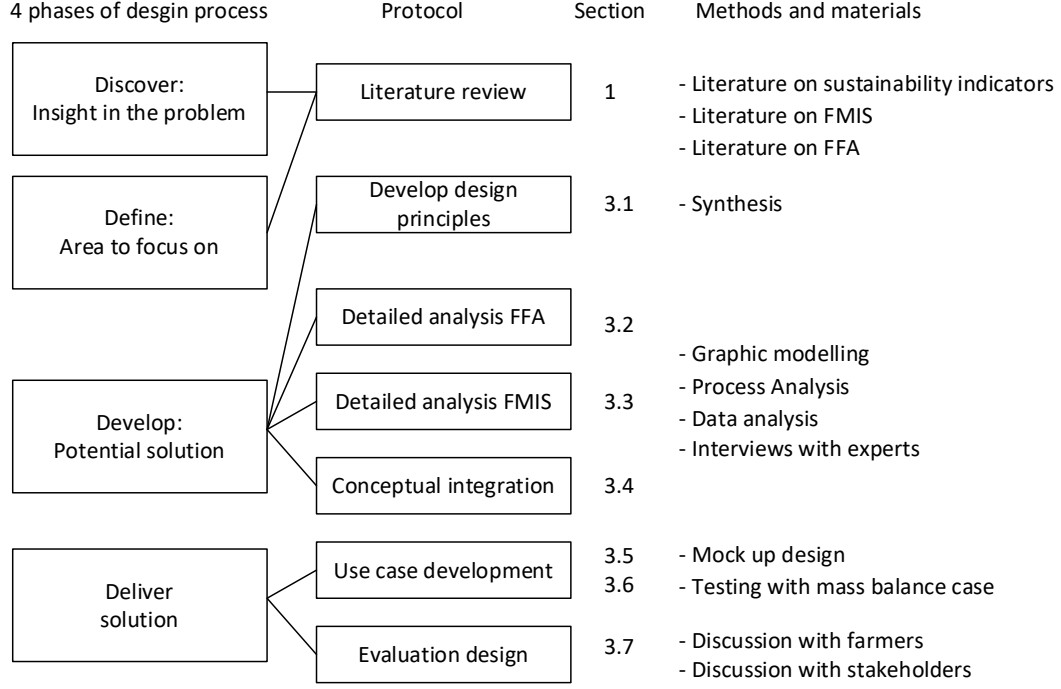

**Figure 1.** Outline of the study: design phases, protocol, and detailed methods and materials.

The design artifact developed in this paper is a conceptual design of a workflow and farmer's dashboard that reduces administrative burdens in external sustainability reporting by integrating information flows from farm accounting systems and farm management systems. Our insights into the problem (Discover phase) and the area to focus on (Define phase), as reported in the Introduction Section, have resulted in a number of design principles that will be described in Section 3.1. In the remainder of the paper, we focus on the phases of developing and delivering solutions.

Based on the relevant literature on the design and use of FFAs and FMISs, we first analyze for each of them how they work: what type of data are entered into a database and from which documents that information is taken. We use the normal administrative business processes (ordering, delivering, using, etc.) as a reference [47].

Given the increasing need for the empirical measurement of sustainability indicators and the characteristics of FFAs and FMISs, we then design an integration of these processes in one coherent workflow and data set and illustrate with a use case how the interaction between the software and the farmer in an integrated workflow in the farm office might work and how a dashboard with sustainability indicators (KPI) is generated.

The material used in this design approach is rather diverse. We used process and data analysis to characterize FFAs and FMISs based on experiences in developing such software. In addition, we interviewed experts in such software and discussed a mock-up of the created artifact with three organic farmers. We looked into the information needs of the Dutch organic certification body concerning mass balances and tested if our design could deliver that statement in conjunction with Value-Added-Tax reporting. Graphical modeling was used to analyze FFAs and FMISs and to create a design for integration. The findings have been discussed with commercial accounting offices and research partners in the European MEF4CAP project.

## 3. Results

### 3.1. Design Principles

The insights from the Discover and Define phase resulted in three design principles. First, the design should lead to less data entry by farmers. In the current situation, invoice information is (manually) entered into payment and financial accounting software as well as in farm management information systems: an invoice for fertilizer is used in making a payment by bank (entering at least the euro amount and an invoice number) and in (linked) farm financial accounting software that creates a Value-Added Tax (VAT) statement (at least the euro amount with and without VAT). In the farm management information system, fertilizer use is documented (the amount of kg used and often also its price or value). Both systems tend to give options to the farmer to record more data (name of the supplier, specific data on the type, and composition of the fertilizer), etc. In some cases, the allocation to a business unit/enterprise (a crop, type of animal), which is recorded in the FMIS, can be done more or less automatically with the invoice data: sugar beet seeds are a cost to the sugar beet enterprise, certain pesticides are only used on specific crops, etc.

A second design principle is that confusing differences between the FFAs and the FMISs, for example in output disappear, and the quality of FMISs data may improve. In the current situation, there tend to be differences in gross margins per enterprise calculated from the FFAs and the FMISs. A reason is that the FMISs often work with standard prices, or the farmer uses the price from a transaction on the invoice. However, in reality, there might be price deductions at the end of an invoice or on an annual basis for bulk ordering, fast payment, cooperative membership, profit sharing, etc. Many farmers also do not record their sales of stored products in an FMIS but stop recording the yield and estimated price. Such differences can be annoying or even lead to incorrect data conclusions.

The third and probably most important design principle is that an integration of the FFA and the FMIS in the farm office should support sustainability accounting and solve a discussion on the choice of the source for environmental indicators and the data for the certification of farms. Examples of such key performance indicators (KPI) are the use of antibiotics, pesticides, and fertilizer, farm gate material balances for nitrogen and phosphate, as well as mass balances that are used in the certification of organic farms and energy inputs that can be used to calculate greenhouse gas emissions. In an earlier paper [22], we argued that FFAs are the best choice for adding sustainability indicators: in a situation where FFAs are obliged for all farmers and FMISs are only used by some farmers, such indicators "could be generated for all farms integrated in the market economy and especially those that are already obliged to keep books for income tax or VAT purposes". This encompasses a much wider group of farmers than the number using a Farm Management Information System and would thus reduce the administrative burden for many farmers. Under the proposed approach, farmers would not be obliged to adopt an FMIS or enter their farm data on a food processor's website.

We also pointed out that data in FFAs can be audited much more easily than in an FMIS [12,48]. Although nearly all farmers will record their data honestly, mistakes and fraudulent misreporting are not unthinkable, especially if good environmental performance is rewarded with support payments and/or higher output prices (as is the case with eco-schemes, organic products, or products with environmental labels). FFAs have methods to verify the completeness of their dataset. "The use of bank account statements guarantees that all payments have been recorded. By linking invoices to these payments, there is some assurance that no invoices have been 'forgotten': if some of the invoices that indicate a lower sustainability level of the farm (e.g., on pesticides) would not have been recorded, then this would not show up as a deductible cost in VAT and income tax statements" [22].

Although the FFAs should be the basis for auditing sustainability performance, there is an extra value in integrating FMISs and FFAs for calculating sustainability KPI. More detailed information on the use of the inputs and their efficiency would become available, including an allocation to crops, fields, and (individual) animals. The allocated data are especially important on multi-crop farms where different food processors (e.g., sugar beet

factory, potato processor, and cereal trader) demand sustainability data on the products they source from a farmer. In such situations, sustainability performance can also depend on the location of a field, the timing of an operation, and the use of the field in previous years for other crops; FMISs have a geographical information system component that can provide such supplementary data. Such integration would also make the FMISs data auditable in a certification process and solve the problem of the potential unreliability of current FMISs data. This reduces transaction costs. The same applies to sensor data which are increasingly integrated into FMISs. With smart farming and its precision farming technologies, big data approaches also become important in agriculture. In the next step of the design, we show how data from sensors and satellites, which are increasingly used in FMISs, can be an additional source of information for sustainability reporting and how the designed dashboard can be responsive to the big data trend.

### 3.2. Farm Financial Accounting

Farm financial accounting uses financial transactions (payment data) to calculate financial statements such as a profit and loss account, balance sheet, and cash flow statement. These are mainly for tax purposes and financial management, for instance, to report to a bank. FFAs focus on monetary flows (euro amounts) with trade partners and on assets. Financial and tax statements are mainly created after the closing of the accounting year. Value-Added Tax (VAT) statements are mandatory on a monthly or quarterly basis, implying that payment data are entered into an accounting system much more regularly. It is the closing of the accounts at the end of the year and fiscal advice that takes time, especially if commercial accounting offices spread their work over the year. This makes the financial accounting information not very timely for farm management, benchmarking in study groups of farmers, or providing data to certification audits unless the basic farm accounting is carried out in the farm office on a monthly or quarterly basis in coherence with the VAT declarations.

FFAs only document a few business processes (Figure 2 gives our graphical model). Offers from suppliers or buyers, orders, and deliveries are not systematically tracked by farmers, with some exceptions like the movement (delivery) of animals and manure, as this is mandatory by European law. The quantities of inputs and output produced (harvest, growth of animals) are also not systematically recorded. Thus, FFAs record payments and related invoices in more or less one process (without documenting the timing difference between receiving an invoice and the subsequent payment as "accounts payable" or "accounts receivable"). Essentially this is a form of single-entry accounting, with a recording of the inventories (and accounts payable/receivable) at the yearly closing of the accounts to calculate the profit or loss and (change in) capital. At that stage, the depreciation of fixed assets is also calculated.

In this accounting process, the costs are based on the use of the inputs as derived from the inputs bought and corrected for inventory change. In a similar way, the revenue (turnover) is based on the output sold, corrected for inventory change. That this is not necessarily equivalent to the production of the farm can be illustrated with products such as potatoes that are stored for some time on the farm: the volume sold in spring is smaller than the volume harvested in autumn due to dehydration.

On mixed farms that have, e.g., a dairy and an arable business unit, or on arable farms with different crops ("enterprises"), variable inputs are allocated to enterprises (type of animals, type of crops) in order to calculate gross margins (revenues minus direct variable costs). Outputs and some inputs (such as seed for sugar beet) can be allocated without additional information, but some allocations ask for data on the use of the input, with the effect that such management information is often not generated.

FFAs have been strongly supported by digitalization for a long time. Accounting offices use accounting software and so do farmers who have some form of bookkeeping to record invoices and create digital payment instructions to their bank. However, most data entry is done manually by typing over some aspects of an invoice. The digitization of in-

voices in a computer-readable format (UBL, XML, etc.) is seen as a bottleneck. Accounting offices try to solve this by installing software to scan PDFs (see [21] for details). Real digitization of invoices would make robotic accounting possible and not only support financial accounting but also automatically integrate environmental accounting and generate many key performance indicators (see [22] for a detailed description).

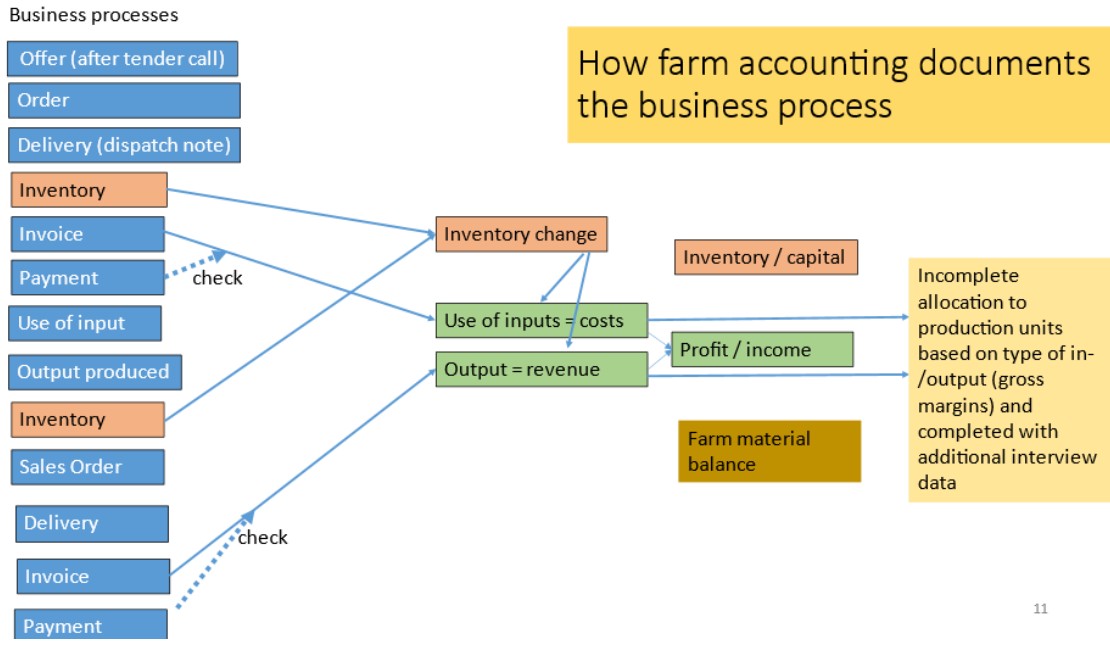

**Figure 2.** How Farm Financial Accounting documents the business process.

*3.3. Farm Management Information System*

An FMIS is essentially a digitalized field book in which farmers record operations. It is a form of management accounting intended to document and guide operational and tactical management decisions. An operation is a certain type of activity (e.g., ploughing, planting, spraying, harvesting, etc.) at a certain field (or a certain (group of) animal(s)), carried out with certain machinery and workers and the use of inputs (seed, fertilizer, pesticides, fuel, etc.) or the harvest of certain product (tons of cereals, potatoes, milk yield) at a certain moment (or during a certain time period). The fields are geo-located, and integration with machinery data leads to more detailed geographically allocated data. This means that FMISs concentrate on material flows in quantities of inputs and outputs, especially on flows within the farm.

Sometimes these operations include operations on delivery of inputs to the farm (inventory) or from the farm (storage) to buyers. This facilitates farmers who have to document animal movements or manure transport. However, many crop farmers close the annual FMIS after the harvest (with an estimated yield) and do not record real sales after storage. Material flows are central in an FMIS, but prices (often the prices taken from invoices without caring much for discounts or different prices from different deliveries) are added to make calculations of gross margins per field, crop or animal enterprise.

Most of the data in an FMIS are typed in manually, but in some situations, data are taken from digital invoices or delivery notes of suppliers. In accounting terms, an FMIS documents the use of inputs per production or business unit ("enterprise"), as well as the output (our graphical model is in Figure 3). Inventory changes can also be supported as part of the recording. The input and output data and their allocation make calculating material balances and gross margins possible. These can be aggregated to the farm level, but it is not really possible to generate a full profit and loss account of the farm as that would require a better recording of a lot of fixed costs (e.g., communication costs, accountancy fees, etc.) as well as correct handling of discounts in prices.

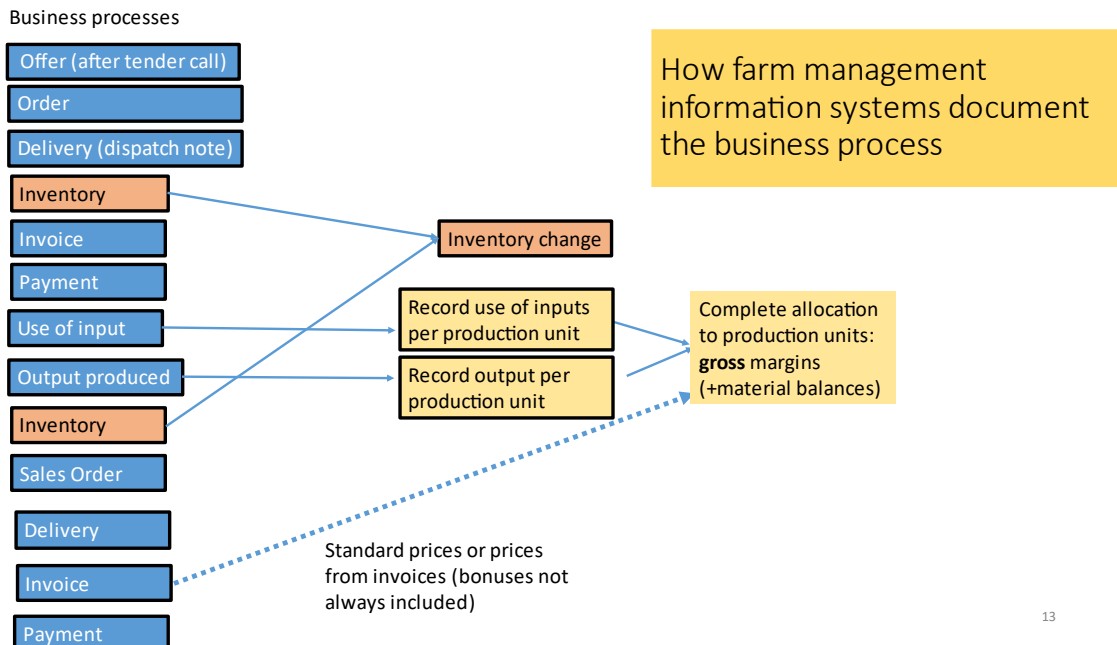

**Figure 3.** How Farm Management Information Systems document the business process.

FMISs are strongly connected to geographical information systems (GISs). Larger farms have several fields and farmers want to follow fields and their cropping plan through the years. The application of subsidies in the Common Agricultural Policy also requires the uploading of field maps. These GIS modules have become increasingly complex over the years as different subsidy arrangements have different demarcations of a field: with or without ditches (for some nature management contracts, the angle of inclination is relevant), with or without farm roads, etc. In other words, fields (or better: temporary field-crop combinations) can have different sizes depending on the use of the data—which is reflected by different coding schemes used for crops and the type of fields by authorities.

Another development in FMISs is the integration with sensor data from machinery used in precision agriculture. This works in two ways. Data on the use of inputs or the harvest of products can be taken from the sensors in the machines with a very detailed (near-point) location on the field maps and aggregation to the field level. This replaces manual data recording. At the same time, FMISs provide options to create prescription maps (task instruction codes to operate the machines) at a near-point location in the field, for example, a variable rate application of pesticides or nutrients.

Although these developments sketch the innovation and future of FMISs in precision agriculture and a big data environment, in reality, most farmers use them not as a decision support system but as a recording tool. Partly for internal purposes to be able to check past operations on crops, animals, and fields, but most of all to report their performance to food processors and their certification schemes with their auditors.

### 3.4. Conceptual Integration of FFAs and FMISs

Integration of Farm Financial Accounting and Farm Management Information Systems without installing a full ERP (enterprise resource planning program) on a farm would be a combination of recording (bank) payments, invoices, the use of inputs with their allocation to enterprises, and the production of output. Such an approach combines the current approaches of the two software applications, uses their strengths, removes redundancy in manual data input, and creates synergies in data quality. The integration would reflect Luca Pacioli's original day book in the farm office, leaving more complicated accounting and auditing tasks to professionals.

The integration of FMISs and FFAs starts with recording the payments and invoices, in which robotic accounting can be used to minimize manual data entry and coding [12].

This creates a register (daybook in accounting terms) of bought/sold with (presumably) delivered inputs and outputs. Where needed (e.g., animals), delivery dates and data can also be recorded, but these are exceptions as a farm does not need a full ERP with data on orders and deliveries or a real-time inventory data set. To integrate this with the recording of farm activities and their use of inputs or outputs and allocation to enterprises, procedures as in an FMISs create a register (daybook) with, respectively, input use and output created (harvest data). Although this remains a manual exercise (if not replaced by sensor data), it becomes easier and provides higher-quality data as a result of integration: it can be supported by suggesting inputs already recorded from the invoices and price data do not to be added. However, this is not a perfect improvement as invoices can arrive after delivery and use and are not necessarily already in the system. Prices are added to the use-data at the moment invoices are available on a FIFO (First in-First out) basis, also taking discounts into account (and until that moment, the software might use standard prices if needed for e.g., planning purposes). Allocation of inputs should include the option to allocate to general use (e.g., maintenance of ditches and farm roads) and an option to label for private consumption (Figure 4 shows our graphical model).

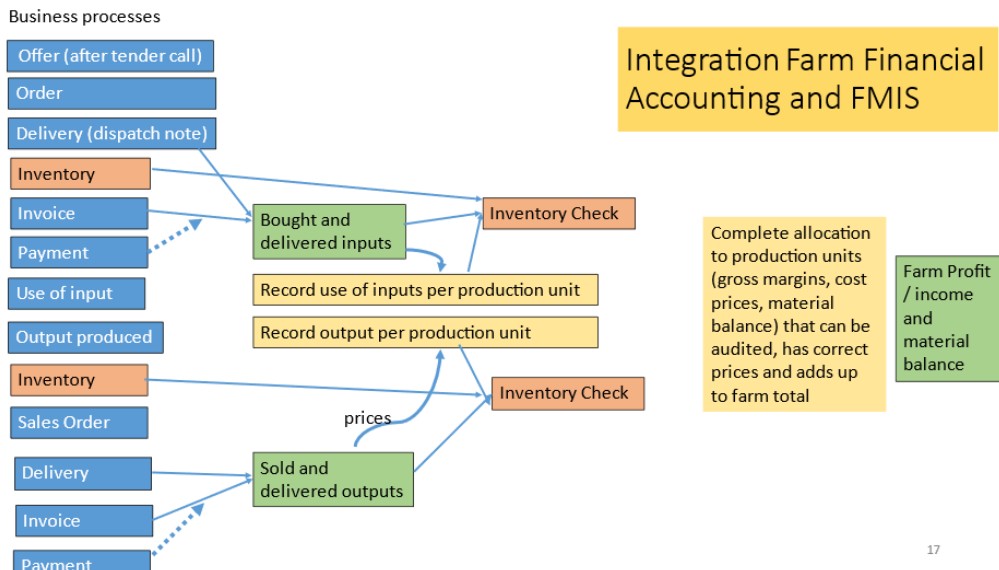

**Figure 4.** How Farm Financial Accounting and Farm Management Information Systems can document the business process in an integrated way.

Inventory data are a check on the recorded transactions and allocations. The end inventory can be calculated as a material (or mass) balance with the bought and delivered amount of a product (from the robotic farm accounting procedures) versus the used/produced amount (from the FMISs procedures). In that procedure, the software user can be prompted to check the real inventory. To handle differences between the actual inventory and the calculated inventory, the user can either enter a forgotten record on user data, add invoices, add a correction record (spilled, evaporated, or lost), or split the difference over the existing user data (e.g., a small difference in the use of a fertilizer that is sprayed many times on different crops is modified with a small difference, for instance, 2.3%).

This integration does not (yet) allow for full cost price calculations. That would ask for the allocation of inputs to individual machinery and buildings, which are treated as an internal contracting (or storage) business unit of which the costs are allocated to crops and animal enterprises. That makes it also possible to handle storage losses: the difference between the harvested volumes of products and the volumes sold and delivered (but without the hassle to track physical and economic inventories as in an ERP).

*3.5. Use Case on Integration of FFAs and FMISs*

In this section, we show how the general approach for integrating FFAs and FMISs, as designed above, would work out in practice in day-to-day use of the computer software—the design of the user interface in the farm office. We imagine an arable farm with several crops and fields. We assume that the software runs on a platform that provides links with bank accounts and informed consent data platforms to exchange digital invoices (and other data). We also assume that the platform has a shared service for computerized coding and maintenance of standard crop and product codes for data exchange and keeps those codes coherent with the demands of government schemes (CAP-IACS), certification schemes (mass balances), etc. In such a platform, an implementation by an individual farm can easily be set up and maintained. We name this software SITRA (System for Information Transfer to Reduce Administrative burdens in the agri-food sector) as proposed in [12]. We come back to these assumptions in a later section of the paper.

We imagine the computer-user interaction as follows:

- Farmer starts up SITRA from his desktop:
    - SITRA shows 6 tabs: Cropping Plan, Finance, Fieldwork, Inventory Check, Asset Management, Statements;
- Farmer selects Cropping Plan tab:
    - SITRA shows the fields of a farmer in a GIS format;
    - SITRA shows a standard list of crops (including an option to handle very specific crops that are not on the list);
- Farmer picks the name of a crop, links it to a field, and enters a starting date:
    - SITRA records this field-crop combination (and the end date, but that can also be derived from the next crop on the field; FMISs have detailed procedures to create cropping plans that fit here);
- Farmer selects the tab Finance:
    - SITRA connects to the farmer's bank account to download recent bank payments;
    - SITRA connects to relevant, informed consent data platforms to download digital invoices in UBL or XML format (and pdf to store for human reading);
    - SITRA checks mail accounts for digital invoices in UBL or XML format and PDF, sent to the farm directly;
    - SITRA matches invoices and bank payments, especially invoices created by clients (food processors, traders) for sold farm produce and invoices for bought inputs where the farmer has given authorization for automatic money withdrawal or that have been paid "cash" with a debit card. These invoices are coded with computerized accounting and a standard chart of accounts;
    - SITRA codes payments with robotic accounting for which invoices are not expected, e.g., installments of payments for communication services or electricity.
    - SITRA presents unpaid invoices to be checked for acceptance;
- Farmer accepts (or declines) the invoice and fills in the payment date(s) for the bank payment instruction (default = today):
    - SITRA creates bank payment instructions and uploads this with the bank. It saves the link between the invoice and bank payment and codes the invoice with robotic accounting;
    - SITRA presents an option to type in invoices that have been received only in paper or PDF form (which are then handled in the way described above, with a mail sent to the trade partner with the request to send the invoice next time also in XML/UBL). The PDF is stored to be available for audits;
    - SITRA presents an option to create invoices by the farmer for occasional sales.
- Farmer selects the Fieldwork tab:

- SITRA displays a screen in which the farmer records an operation, selecting from a drop list a certain type of activity (e.g., ploughing, planting, spraying, harvesting, etc.) and a certain field (of which the crop is known via the cropping plan) as well as the machinery, labor, and inputs (seed, fertilizer, pesticides, fuel, etc.) used or products harvested. The inputs are selected from products on invoices (or also delivery notes if these are also handled by SITRA under the tab Finance and linked to invoices);

- Farmer records the operation. If the input used is not presented on the basis of the invoiced products, he selects a standard list of products and makes a choice;
- Farmer selects the tab Inventory Check:
    - SITRA calculates for all products (inputs, outputs) the theoretical inventory based on the formula: bought (invoices) minus used (fieldwork operation) or harvested (fieldwork operation) minus sold (invoices);
    - SITRA provides the farmer with an opportunity to fill in the actual inventory at that date and calculates a difference;
    - SITRA offers options to handle differences between the actual inventory and the calculated inventory: entry of a forgotten record on user data, add invoices or split the difference over the existing user data (e.g., a small difference in the use of a fertilizer that is sprayed on different crops is modified with a percentage);
- Farmer improves data quality by supplementing extra data to make the theoretical and real inventory coherent;
- Farmer selects the Asset Management tab:
    - SITRA provides the option to update fields (selling, buying, renting in or out), machinery (selling or buying with data entry on expected remaining lifetime and scrap value), buildings (ibid), and quota (ibid);
- Farmer selects an option and fills in the required data;
- Farmer selects the Statements tab:
    - SITRA runs an audit module that checks the validity of data (by assessing if data such as prices or use per ha are within a certain confidence interval and some other logic checks). Includes a check if the inventory check (above) has recently been carried out. Strange values are presented for confirmation;
- Farmer confirms value as being true or corrects earlier data:
    - SITRA calculates mass balances for products that show the relationship between bought inputs (especially seed) and outputs sold plus still in stock;
    - SITRA calculates material balances for Nitrate, Phosphate, Potassium, Pesticides, Antibiotics, and Greenhouse gases;
    - SITRA calculates a cash-flow statement;
    - SITRA calculates gross margins per crop/field (using standard prices for those inputs where according to a FIFO calculation, some invoices still have to be received);
    - SITRA calculates a profit and loss account, income statement, balance sheet, and flow-of-funds statement in case assets have also been recorded;
    - SITRA calculates the cost-price statement per crop/field if machinery use has been recorded in the Fieldwork tab, and costs can be calculated from the Asset Management data;
    - SITRA calculates Key Performance Indicators in economic and environmental sustainability;
    - SITRA generates statements for the application of subsidies in the Common Agricultural Policy (based on field maps), certification audits, and a daybook or journal for the accounting offices to create tax accounts;
- Farmer selects relevant statements for study or distribution to external partners. Adds information where needed.

This example for crop farming can easily be adapted for animal production, where the type of animals (dairy, pigs, etc.) or individual dairy cows are the business units/enterprises.

Often such farms also have fields (grassland) that can be seen as equivalent to the crops on an arable farm.

In the Fieldwork tab, the operations should include the option to allocate inputs to general use (e.g., maintenance of ditches and farm roads) and an option to label for private consumption. A more advanced version, especially for cost price calculations, could include the allocation of inputs to individual machinery and buildings, which are then allocated as an internal contracting (or storage) business unit to crops and animal enterprises. That also enables handling storage losses: the difference between the harvested volumes of products and the volumes sold and delivered (but without tracking physical and economic inventories as in an ERP). In more advanced systems, there will be a planning module in the software, and the recording can be based on a change of the status from "planned" to "performed" and an update of those data.

The SITRA—farmer interaction can be more detailed based on the best practices in the current FFAs and FMISs. The example is robust because, in some situations, the use of inputs in operation preludes the reception of an invoice for those inputs. With robotic accounting and a tendency to send invoices by email very quickly after ordering and delivering (some companies already provide an app to farmers that support the full process of demanding a quote, ordering, delivering, invoicing, and paying as in an ERP system) this will be less often the case than in a world with paper flows.

### 3.6. The Role of Sensor Data in the Integration of FFAs and FMISs

The trend in digitalization of the Internet of Things implies that more and more data is becoming available on the use of inputs or harvest of outputs from sources like sensors, actuators in tractors, machines, (milking) robots, drones, and satellites. Examples include sensors that constantly measure the amount of feed in a silo and trigger new orders and delivery when needed. Other sensors in precision farming measure the uptake of feed by individual animals, even including an estimation of grass eaten in the meadow. Sensors in precision farming measure the spraying of fertilizer or pesticides. An example is an app that scans the package of pesticides (bar code) in the field, uses the field size from the FMISs, downloads the weather forecast and advice on how much pesticide to use and how much water to add, and creates a machine instruction to fill up the tank. Sensors in combine harvesters measure the yield of a (part of a) field, and milking robots report yields per individual cow. There are also sensors that measure emissions at the farm (e.g., fine particles, ammonia). After benchmarking with pollution in the neighborhood as background "noise," these data can be used to estimate the emission of the farm and linked to material balances and used in certification processes. Such emissions and potential yields can also be measured with satellites that send their data to a farmer.

This trend to data generated with Internet of Things (IoT) technologies has consequences for the accounting process: data on the application of inputs or the harvest of products (output) does not have to be recorded manually but can be based on the IoT sources, as FMISs already do (Figure 5 shows the graphical model). That means less administrative work and fewer risks of mistakes. IoT data also make it possible to identify production units at a much lower level: the individual animal or tree, parts of a field. This makes it possible to support decision-making with gross margins that can be calculated at that level. A third consequence is that data in material balances of inputs like pesticides, feed, or fertilizer and their derivate material balances of Nitrogen and Phosphate can be extended with emission data, potentially leading to better management of emissions and more tailored public regulation [49,50]. These data can, best of all, be linked to the processes of registering inputs and outputs in operations that are currently in the FMISs not only because they are used in the first place in operating these processes but also because they mainly provide data on use and production and, for interpretation, can be linked with weather data at the moment of the operation.

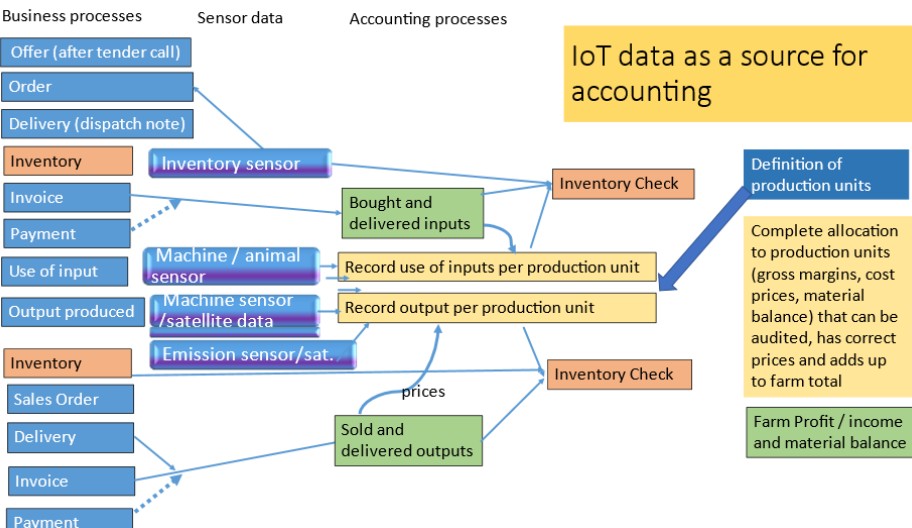

**Figure 5.** How IoT data can be integrated with Farm Financial Accounting and Farm Management Information Systems data. (Some sensors not only register, as the figure suggests, but also act as actuators to steer the business process, as in variable rate application).

*3.7. Data Flows to and from the Farm Office*

The analysis above shows that with current digital innovations (such as UBL, robotic accounting, and data vaults (data spaces) for individual farmers in the cloud), it is possible to integrate FFAs and FMISs and reduce the administrative burden in the farm office. An integrated workflow results in one dashboard (graphical user interface) with financial and management statements with their key performance indicators (and related GIS maps also used in IACS) as well as sensor and satellite data without installing a full ERP system.

As explained by [21] in Figure 6, such software in the farm office benefits from informed consent data platforms for handling digital messages with authorizations and identification procedures and a shared coding service that supports rules for robotic accounting and uniform coding. This shared coding service would guarantee that, e.g., crop codes are equivalent to those used in CAP subsidy schemes or statements with mass balances are acceptable to certification organizations. These functionalities could be created at a regional or national level as a collective action or public utility or as part of a commercial platform. Based on this common data management, the software for the individual farm office would generate financial accounts, management accounts, and environmental accounts that the farmer could share with their advisers, accountant, and certification bodies. Data could also feed into more complex FMISs or farm decision support tools that interact with machinery.

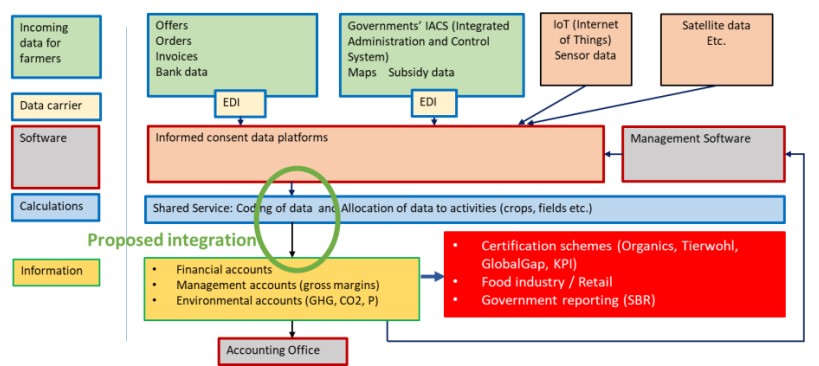

**Figure 6.** Data flows and the place of a Farm Financial Accounting and Farm Management Information Systems integration.

## 4. Discussion

The results of our analyses show that it is possible to design a workflow and dashboard to meet the increasing sustainability reporting needs in a way that minimizes data entry, assures consistency in sustainability indicators between FFAs and FMISs, and allows for the auditability of results. In this section, we will discuss how the design in this paper compares to alternative approaches which have been proposed. The limitations of the design science approach will be addressed. The discussion section ends with the implications for research and policymaking.

An alternative approach to integrating data flows for sustainability reporting is the use of ERP systems. Their use in practice has been concentrated on very large farms; a lighter type of ERP system for family farms with mainly self-employment and less need for internal management information is lacking. The adoption of ERP on family farms is low—a situation comparable to the historical example of the much-advocated double-entry accounting that seldom replaced single-entry accounting in farming [51].

Another alternative approach to meet the increasing need for sustainability reporting is the development of sector-wide central data systems for reporting sustainability in farming with minimum involvement of farmers. Based on the authorizations of farmers, the invoice and other data are brought together in a central database, and KPIs are calculated for the farmer to report the sustainability performance of the sector to the public, government, and clients further down the chain. An example is the ANCA system (annual nutrient cycle assessment) in Dutch dairy farming, which is a follow-up of the mineral accounting system [52]. This approach has the advantage of minimizing the burden on farmers but also has two problems. Like standalone FMISs, the data are hard to audit as they are not linked to financial flows, and perhaps more importantly, by leaving the farmer out of the loop, the data are not grounded or much used in management decisions. For instance, the $CO_2$ footprint of feed is calculated at the end of a year, and feed suppliers do not include that information on the delivery notes or invoices that farmers receive at an individual order earlier in the year. Getting quotes for feed with different $CO_2$ footprints is extremely difficult. This makes it hard for farmers to act upon the information that influences their sustainability performance.

Reviews of design science studies reveal that there is wide variation in the steps and methods applied [45,46]. In our study, we have conducted four steps in the design process and have used various methods in these steps. A critical reflection on our approach shows that our literature review of FFAs and FMISs focuses on papers analyzing the overall functionalities of these systems. Furthermore, our design is based on the situation in the Netherlands. In other countries, the adoption and use of FFAs and FMISs, the need for sustainability reporting, and the amount of data exchange in the agricultural sector might differ. A more detailed review of FFA and FMIS systems used in specific countries or sectors would provide input for a further strengthening of the design. Another limitation is the lack of empirical validation of our artifact. Due to the type of artifact, a design of the integrated approach, we were restricted from evaluating the design based on discussions with farmers and other stakeholders. Full implementation and validation would not have been possible at this stage. First, several obstacles must be solved, as discussed in the next paragraphs. This paper could help to solve these obstacles as it illustrates the advantages and helps to mobilize relevant stakeholders. The obstacles that need to be solved are fear of transparency, the self-interest of food chain partners, and the competencies of companies.

The fear of transparency comes from input suppliers, food processors, and farmers. Companies delivering feed or pesticides to farmers have been reluctant to provide bulk digital invoices to FFAs or FMISs because they fear real-time price comparisons by accounting offices. In the case of an obligatory mineral accounting system in the 1990s [48], a contract was made up to explicitly forbid this type of use.

Farmers might have a similar fear of transparency because they are unsure whether they have respected all different and frequently changing regulations. We came across an example of farmers who want to restrict the data exchange with their food processors to a minimum needed for managing the crop (where, e.g., a sugar company wants to provide

its clients full transparency on the sustainability of the production). For instance, a farmer that has his crop sprayed on a wind-free Sunday night by a 17-year-old farm hand (who has a spraying license) for a low hourly wage can be unsure if this operation is fully in accordance with labor laws and is reluctant to share the full operational data, even if the use of the pesticide is fully in line with the law and best farm practices.

The second obstacle to be solved is the self-interest of stakeholders. For food processors and certification schemes, it is easier to set up a dedicated website where farmers are obliged to provide their data than to invest in a system that also reduces the administrative burden of farmers. In the allocation of IT budgets, the factory and the internal administration have priority over improved digital interaction with suppliers and members of the cooperative. Accounting offices have also concentrated on computerizing internal processes. Not many farmers were interested in coding transactions themselves; the coding had to be checked for quality, and it did not reduce the accounting invoice for farmers. Some accounting offices preferred to do the data entry themselves to achieve a higher margin, although that has changed with shortages in the labor market. Not being able to force input companies and food processors to provide digital invoices, they started experimenting with scanning [21]. In the Netherlands, interest in detailed invoice data and environmental accounting has been low. The business of certification of such data implies a critical attitude that conflicts with the trust relationship they have with farmers in their financial consulting. Neither do they have the expertise in (local) operational and environmental advice. That is left to private extension consultants.

In the past, Dutch banks have tried to upscale their digital payment systems with options to handle invoices and Giro collection forms in an application called FINBOX. This solution (not specific to agriculture) was unsuccessful and was abandoned in 2016/17. The PSD2 regulation has obliged banks to make payment data digitally available to third parties (such as accountants or fintech companies) when authorized by the account holder. This indicated that separated markets for data integration should be developed instead of a central role by banks. In the Netherlands, JoinData was formed by input companies and food processors as an informed consent data platform. It restricted itself to a strict passive data exchange facility, with a focus on precision farming, and refrained from actively promoting the digitization of invoices.

All in all, the self-interest of the usual stakeholders that create or handle invoices and accounting data has not been large enough to create an integrated system as we designed. It was easier for stakeholders to innovate and solve the immediate needs of their own company than to create a collective action for a system that could be more optimal from the farmers' point of view. The mineral accounting system of the 1990s came close to an integrated system in animal husbandry [48] but only under the threat of strict environmental legislation. It immediately stopped when the legal obligation was withdrawn.

The third obstacle that could limit the results of our design study is the need for competencies with farmers and software vendors to work with the software based on the design. As farmers already pay their bills using digital banking systems, digital competencies are probably not a big bottleneck, but the volume of the reports and indicators provided with robotic accounting based on this payment process might scare some of them. Software companies are not guaranteed to have competencies in accounting and farm management processes, and they might be reluctant to invest in this.

### 4.1. Contribution to Science

The contribution of this paper is linking the data needs for sustainability reporting and the data systems available at the farm. Reporting sustainability indicators based on a workflow around the available systems contributes to both the field of sustainability measurement and the field of data management at the farm. This paper advances the scientific knowledge about sustainability measurement. In the (recent) past, many frameworks of indicators have been developed. However, the experience with collecting data for these indicators at the farm level was very limited. This paper contributes to the feasibility of sustainability measurement and,

therefore, to the knowledge and value of these frameworks. By providing a systematic way of measuring sustainability indicators, a broad range of new research projects on trade-offs and jointness between farm-level sustainability aspects become more feasible. The paper advances the knowledge about FFA and FMIS systems by valorizing the data of these systems, providing an additional incentive for farmers to use these systems, and providing directions for future research on the role and operation of these systems.

Our study also provides suggestions for further research. Studying the problems in the implementation and adoption of our designed workflow and dashboard is a logical next step. Furthermore, important questions are the business model, the governance, and the up-front financial investment in the integrated dashboard. Especially in small markets such as the Netherlands, it is not so clear if it is attractive for a vendor of FMISs or FFAs to acquire additional expertise to realise the integration. That could ask for collective action (e.g., by a farmers' organization) or public intervention and will raise questions on the business model (payment by a levy or a user fee) and the governance of the data platform (with different potential roles for stakeholders like data providers, accountancy and certification organizations or food processors and governments that all benefit from the data supply and the farmers themselves).

### 4.2. Contribution to Policy and Practice

The design of an integrated workflow for sustainability reporting has a couple of policy implications as well as for business practices. Governments increasingly want to stimulate and reward farm management that improves sustainability performance, but governments face problems in measuring and monitoring this performance. Governments can benefit from the described approach and help further implement it. Especially the digitization of paper flows is a collaborative undertaking in which governments could take the lead, in a voluntary approach or by obliging digitalization, in a similar way that banks have been obliged under PSD2 to make data digitally available. As discussed above, such innovation is not automatically induced by the market.

In the Farm-to-Fork strategy, the European Commission has proposed to upgrade its Farm Accountancy Data Network into a Farm Sustainability Data Network (FSDN) [7]. How to collect these additional sustainability indicators is still a major unsolved issue. National FSDN could benefit from better use of available information at the farm level and provide farmers with software for data management. Although FFA software based on digital invoices and robotic accounting would already provide the FSDN data at the farm level, it might be interesting to also have data on individual crops and fields and introduce an approach as described in this paper.

A third policy implication relates to how governments collect sustainability data on all farms that have to report to government agencies, e.g., for cross-compliance in the CAP, for organic certification to monitor the execution of eco-schemes, or for CAP conservation contracts. In that case, farmers could upload sustainability indicators to the Integrated Administration and Control System (IACS) system while uploading VAT statements to the fiscal authority. The government should have methods to verify the data, for instance, by demanding access to the underlying data (such as invoices) or by letting a trusted third party (such as an accountant) verify the data integrity. Such an approach would fit in the recommendations of advisory councils [49,53] and researchers [50,54] that governments could use digitalization more in sustainability policies.

Given the low penetration of FMISs (certainly under smaller and medium-sized farms) and on-farm accounting software and the fear of farmers to be fully transparent (given all the complicated and volatile regulations as discussed above), one might be skeptical about such an approach that could easily be portrayed as the "Big Brother from Brussels". However, the idea could not be so far-fetched if embedded in a certification approach such as in organic farming. In the certification of organic farms, the farms already have to supply mass balances (to show the relation between inputs and outputs as an indication of the use of organic inputs) and open up their administration during a farm inspection.

The design of an integrated workflow for sustainability reporting has similar relevance for the private sector, as many food chain companies have an interest in farm data. Many farms are certified under food safety or sustainability schemes in the food chain, which probably increases further due to the CSRD-scope3 obligation. Food chain companies could help farmers and software companies in realizing the design, e.g., by making their invoices digital and benefiting themselves from better data. It is also plausible that the same certification could be used to monitor CAP arrangements [53,55]. Inspections of certified farms could easily take into account high-quality datasets of those farmers who have such data and use other additional inspection processes for farmers who do not use sensors or integrated FFAs-FMISs software. In such an approach, there are incentives for farmers to adopt integrated FFAs-FMISs software as it reduces their administrative burden towards food processors and the government in certification. The government would be incentivized to oblige food chain partners to introduce digital invoices and promote the software to support learning processes on sustainability issues with farmers.

## 5. Conclusions

In this paper, we showed how the increasing need for sustainability reporting could be met by better use and integration of data flows at the farm. A dashboard with key performance indicators on the sustainability performance of a farm integrates farm financial accounting and farm management information. FFAs and FMISs operate on different levels of granularity. FFAs are more complete, while FMISs provide far more detail. Integration leads to an easy-to-use digital dashboard in which data are coherent. This improves the auditability of FMIS data and reduces manual entry. Sensor and satellite data can be added to further improve the data quality and reduce manual data input.

Such a dashboard has the potential to integrate sustainability considerations in day-to-day management decisions and is therefore preferred to approaches that mainly document the farm's sustainability performance in central databases instead of in on-farm decision support systems. Key performance indicators would become available in the same process as generating monthly or quarterly VAT statements.

This approach could be useful for monitoring the CAP and control systems for organic farming. Several aspects of the contracts or conditionality requirements of these policies are not observable from outside the farm. It is the purpose of the policies that sustainability aspects are integrated into farm management which makes the calculation of sustainability KPI in the management software attractive. The integrated FFAs-FMISs approach could be embedded in a certification process for all farmers who benefit from the CAP. These farms are often already in certification schemes of the food industry, and an alignment of indicators, and perhaps even the certification itself, could provide clear sustainability signals to farmers. The integration of FFAs and FMISs reduces the administrative burden for farmers and frees them from the farm office to let them tend their soil and animals.

In future steps, some limitations of our study need to be addressed. Specificities of countries or sectors could lead to minor changes in the design. Furthermore, an empirical evaluation of the proposed approach should provide proof of concept. This requires the resolution of a few obstacles in the sector: the fear of transparency, the self-interest of stakeholders, and the need to increase competencies.

**Author Contributions:** K.P. contributed to the conceptualization, methodology, formal analysis, writing, and visualization. H.V. contributed to conceptualization, funding acquisition, writing, review, and editing. I.B. contributed to the conceptualization, analysis, review, and editing. All authors have read and agreed to the published version of the manuscript.

**Funding:** This project has received funding from the European Union's Horizon 2020 research and innovation program under grant agreement No. 101000662. The content of this publication exclusively reflects the author's view, and the Research Executive Agency and the Commission are not responsible for any use that may be made of the information it contains.

**Acknowledgments:** This paper has been written in the context of the European MEF4CAP project and builds upon a case that aims to realize a System for Information Transfer to Reduce Administrative Burdens (SITRA) in the agri-food sector (Poppe, Vrolijk, and Van Dijk, 2021).

**Conflicts of Interest:** The authors declare no conflict of interest.

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
