# Peer review of "Integration of Farm Financial Accounting and Farm Management Information Systems for Better Sustainability Reporting"

_electronics, doi:10.3390/electronics12061485_

Round 1
Reviewer 1 Report
The paper uses a conceptual approach to design the integration and develops a use case for the 23 farmer – software interaction. This illustrates that if invoices would be digitized, most data gather- 24 ing would automatically be done in the process of paying the invoices by bank. Data on use of inputs 25 and production could be added with procedures as in current FMIS, but with less data entry and 26 less risks of differences in outcomes.
Reviewer 2 Report
Overall, the paper is written well to present a conceptual approach to designing the integration and developing a case for farmer software interaction. The article explains how digitization enables the payment of invoices to be seamless via the automation of the data-gathering procedure. The paper fits the knowledge of management information systems for improved sustainability reporting. The outcome of the integration makes data entry less cumbersome. It adds valuable data to the environmental account, though there is less evidence of data mining but a high level of conceptualization.
The result provides evidence that the current digital innovation enables the effective integration of FFA and FMIS for improved reporting and reduces the administrative burden for farmers, contributing to knowledge.
Reviewer 3 Report
The paper analyzes the integration process between two information systems used in agricultural companies. The paper does not have a scientific slant, rather an operational one. It does not contribute scientifically to the improvement of scientific knowledge. For companies, including agricultural ones, systems called ERP (enterprise resources planning) have already been developed, which are integrated systems.
Reviewer 4 Report
1. Authors should provide evidence from literature indicating the research gap their work is filling.
2. What methodology was used in the study and how was it described? Was the methodology adopted or adapted from previous studies? A clear step by step protocol used in the study should be specified.
3. The methodology is titled 'Materials and Methods'. The materials used in the study were not indicated.
4. What are the contributions of the study to theory and practice?
5. The limitations of the study were not stated.
6. The directions for future studies were not specified.
7. In the discussion, the results were not compared and/ or contrasted with findings from prior related studies.
Round 2
Reviewer 3 Report
thanks to the authors for the changes, the approach adopted and principles of design science have improved the paper.
I recommend specifying the limits of the paper in the conclusions and in particular the problems of the companies in the sector regarding the skills and resistance to implementation.
Reviewer 4 Report
1. Authors should clarify what they mean by "Integration of FFA with FFA" in the third paragraph of the abstract.
2. Authors should cite supportive references to support the research gap their work is filling.
3. The design science methodology has a detailed protocol. Authors should provide and follow the protocol in explaining their research process and also indicate how their study aligned with the methodology and with the research objectives. Authors should explain step by step, how design science methodology and protocol were used in the study.
4. Any contribution of the study to practice and to industry?
5. How does the study findings compare and /or contrast with the findings from prior related studies?
Round 3
Reviewer 4 Report
1. The abstract did not contain the research gap for the study.
2. Some spelling mistakes are observed, for example, on line 74, etc.
Author Response
Dear Reviewer,
We have added one sentence to the abstract to describe the research gap.
We have corrected a few spelling mistakes.
kind regards, the Authors